# Enhancing Lithium and Sodium Storage Properties of TiO_2_(B) Nanobelts by Doping with Nickel and Zinc

**DOI:** 10.3390/nano11071703

**Published:** 2021-06-28

**Authors:** Denis P. Opra, Sergey V. Gnedenkov, Sergey L. Sinebryukhov, Andrey V. Gerasimenko, Albert M. Ziatdinov, Alexander A. Sokolov, Anatoly B. Podgorbunsky, Alexander Yu. Ustinov, Valery G. Kuryavyi, Vitaly Yu. Mayorov, Ivan A. Tkachenko, Valentin I. Sergienko

**Affiliations:** Institute of Chemistry, Far Eastern Branch of the Russian Academy of Sciences, 690022 Vladivostok, Russia; svg21@hotmail.com (S.V.G.); sls@ich.dvo.ru (S.L.S.); gerasimenko@ich.dvo.ru (A.V.G.); ziatdinov@ich.dvo.ru (A.M.Z.); alexsokol90@mail.ru (A.A.S.); pab@ich.dvo.ru (A.B.P.); all_vl@mail.ru (A.Y.U.); kvg@ich.dvo.ru (V.G.K.); 024205@inbox.ru (V.Y.M.); tkachenko@ich.dvo.ru (I.A.T.); sergienkovi@yandex.ru (V.I.S.)

**Keywords:** TiO_2_(B), doping, nanobelts, mesoporosity, lithium-ion battery, sodium storage, anode, safety, electrochemical performance

## Abstract

Nickel- and zinc-doped TiO_2_(B) nanobelts were synthesized using a hydrothermal technique. It was found that the incorporation of 5 at.% Ni into bronze TiO_2_ expanded the unit cell by 4%. Furthermore, Ni dopant induced the 3*d* energy levels within TiO_2_(B) band structure and oxygen defects, narrowing the band gap from 3.28 eV (undoped) to 2.70 eV. Oppositely, Zn entered restrictedly into TiO_2_(B), but nonetheless, improves its electronic properties (*E*_g_ is narrowed to 3.21 eV). The conductivity of nickel- (2.24 × 10^−8^ S·cm^−1^) and zinc-containing (3.29 × 10^−9^ S·cm^−1^) TiO_2_(B) exceeds that of unmodified TiO_2_(B) (1.05 × 10^−10^ S·cm^−1^). When tested for electrochemical storage, nickel-doped mesoporous TiO_2_(B) nanobelts exhibited improved electrochemical performance. For lithium batteries, a reversible capacity of 173 mAh·g^−1^ was reached after 100 cycles at the current load of 50 mA·g^−1^, whereas, for unmodified and Zn-doped samples, around 140 and 151 mAh·g^−1^ was obtained. Moreover, Ni doping enhanced the rate capability of TiO_2_(B) nanobelts (104 mAh·g^−1^ at a current density of 1.8 A·g^−1^). In terms of sodium storage, nickel-doped TiO_2_(B) nanobelts exhibited improved cycling with a stabilized reversible capacity of 97 mAh·g^−1^ over 50 cycles at the current load of 35 mA·g^−1^.

## 1. Introduction

Up until recently, electrochemical energy storage devices, among which lithium-ion batteries (LIBs) are dominant, were mainly used for areas that demanded moderate characteristics and soft operating standards (e.g., portable electronics, medical equipment, and power tools). At present, they are closely considered for usage in hybrid and electric vehicles, renewable energy, robotics, and standby backup power applications. However, due to the fundamental physicochemical properties of carbonaceous anode-active material (the voltage of Li^+^ insertion is close to that of Li metal formation, thereby causing its dendritic deposition; this is critical for charge in high-rate or low-temperature conditions), LIBs suffers restricted performance for such aims, especially in terms of charging rate, operating temperature range, and safety [1,2,3]. To avoid these problems, a negative electrode based on lithium pentatitanate with an operating potential of 1.55 V vs. Li/Li^+^ [4] was recently proposed and successfully commercialized (for example, in Mitsubishi i-MiEV and Honda Fit EV electric cars). Another important advantage of Li_4_Ti_5_O_12_ is cycle life upon lithiation/delithiation due to its so-called “zero-strain” property. At the same time, the specific capacity of Li_4_Ti_5_O_12_ is limited to 175 mAh·g^−1^ [5]; hence, designing other materials, in which more Li can be accumulated, is needed to achieve further progress in terms of the charge rate and temperature range of LIBs. One such material can be titanium dioxide, which possesses an Li-storage capacity of up to 335 mAh·g^−1^ [6] at a similar Li^+^ insertion voltage of 1.5–1.8 V vs. Li/Li^+^ (depending on the crystal structure). In this way, layered structural monoclinic (space group *C*2/*m*) bronze titanium dioxide (TiO_2_(B)) showed the most favorable performance as an LIB anode in comparison with other natural polymorphs [7,8]. The TiO_2_(B) framework belongs to ReO_3_-type structure [9,10], which is presented by packings of ReO_3_ blocks composed of distorted TiO_6_ octahedra sharing the corners. The linking of blocks in the [001] direction is achieved through common edges. The packing of ReO_3_ blocks in the [010] direction is displaced to a neighboring one by a/2 (a is unit cell parameter) along the [100] direction. Therefore, TiO_2_(B) has an open structure with infinite channels running parallel to the *b*-axis. Such structural features govern the lithium insertion into bronze TiO_2_ as a pseudocapacitive faradaic process [11], which is not hampered by sluggish solid-state diffusion unlike anatase, rutile, or brookite. Moreover, TiO_2_(B) has a lower Li^+^ insertion potential (near 1.5 V) compared to other natural polymorphs, resulting in higher energy density of the battery.

On the other hand, lithium reserves in the Earth’s crust are limited, and the cost of lithium-containing raw materials is growing steadily. In this way, a change to other electrochemical energy storage technologies based on abundant materials, such as sodium-ion batteries (SIBs), is expected in the near future, at least for renewable energy and uninterruptible power supply fields [12,13]. At the same time, due to the unsuitability of graphite, other materials are being studied intensively as an anode for SIBs [14,15]. In view of the background and progress regarding LIBs over the past three decades, it may be postulated that intercalation-type materials are the most valuable for these purposes. However, as known, one of the main differences between SIBs and LIBs is the radius of charge carrier (1.02 Å for Na^+^ and 0.76 Å for Li^+^), which ensure an electrochemical process. Therefore, the search for materials with acceptable stability of structure (preferably with layered or tunnel crystal frameworks) under cyclic Na^+^ insertion and extraction is an urgent task. Furthermore, the near-zero potential vs. Na/Na^+^ of a hypothetical anode material may result in metallic sodium deposition on its surface, which limits battery operation and causes safety problems. Due to these reasons, titanium compounds based on the Ti^3+^/Ti^4+^ redox couple, including NaTiO_2_, Na_2_Ti_3_O_7_, TiO_2_, Na_2_Ti_6_O_13_, Na_2_Ti_4_O_9_, NaTi_2_(PO_4_)_3_, and ATiOPO_4_ (A = NH_4_, K, Na), have attracted attention [16,17]. Regarding four common TiO_2_ polymorphs, it was found that bronze titanium dioxide possesses the best sodium storage properties [18,19]. The potential of TiO_2_(B) vs. Na/Na^+^ is 0.6–0.9 V [20], making it suitable as an SIB anode in terms of both energy density and safety concerns.

Unfortunately, TiO_2_(B) is a wide-band gap semiconductor (its calculated band gap energy ranges between 3.09 and 3.22 eV [11,21]) with low electrical conductivity (~10^−12^ S·cm^−1^). In addition, the cyclic performance of TiO_2_(B), depending on its volume variations upon the insertion and extraction of guest ions, especially in the case of accommodating large Na^+^, needs to be improved.

Up to date, a number of strategies have been proposed to address these issues, such as morphology tailoring, fabrication of composites, and metal and/or nonmetal doping. It has been revealed that success necessitates a combination of approaches involving microstructure tuning and doping routes. Indeed, when TiO_2_(B) is reduced to nanoscale, e.g., nanotubes [22,23], nanorods [24], nanofibers [25], nanoribbons [26], nanowires [27,28], or nanoplates [29], it usually shows enhancing rate properties as an anode. Moreover, special attention should be paid to the mesoporosity of nanomaterials, which effectively ensures electrolyte penetration and charge carrier transport. On the other hand, doping titanium dioxide with metals/nonmetals [30,31,32,33] further increases its electrical conductivity due to the formation of local energy states inside the band structure and/or the generation of lattice defects (oxygen vacancies, Ti^3+^ species). Furthermore, controlling the variation in the radius of host and doped ions [34,35], as well as supporting the oxygen deficiency [36], may provide an increased stability and activity of TiO_2_(B) during cycling in SIBs.

Herein, we successfully synthesized mesoporous TiO_2_(B) nanobelts doped with nickel and zinc through a hydrothermal reaction. It is expected that, in terms of lithium and sodium battery performance, Ni and Zn may be suitable dopants for TiO_2_(B), providing both enhanced cyclability and rate capability. As suggested, the main factors determining such improved electrochemical behavior of doped TiO_2_(B) are (i) increased conductivity of the material, (ii) improved stability of its structure during insertion/extraction of guest ions, and (iii) facilitated diffusion of charge carriers.

## 2. Materials and Methods

### 2.1. Synthetic Procedure

For preparing mesoporous nanobelts of doped TiO_2_(B), anatase with an average particle size of ~100 nm (Alfa Aesar, Ward Hill, MA, USA), nickel and zinc nitrate hexahydrates (Merck, Darmstadt, Germany) and aqueous solutions of sodium hydroxide (14 M) and hydrochloric acid (0.05 M) were used as precursors. In brief, 0.1 g of anatase was dispersed in 15 mL of 14 M NaOH under vigorous stirring on a magnetic stirrer for 20 min. Then, a weighed portion of Ni(NO_3_)_2_∙6H_2_O or Zn(NO_3_)_2_∙6H_2_O was added in the solution in order to obtain the following atomic ratios of doped metal to titanium: 0.02, 0.05, and 0.08. The resultant solution was transferred to a 20 mL Teflon-lined stainless-steel autoclave and heated at 170 °C for 72 h. After the reaction, the product was cooled naturally to room temperature, separated on a 5804R centrifuge (Eppendorf, Hamburg, Germany), rinsed in 0.05 M HCl (three times, 75 mL each) for Na^+^/H^+^ exchange, washed in deionized water until the pH became neutral, and dried in air at 80 °C. Finally, the sample was treated thermally under vacuum (1 Pa) at 450 °C for 3 h to crystallize the bronze phase. The obtained series of materials were designated as TO-Ni-*z* and TO-Zn-*z*, where *z* represents the content of doped metal. The undoped TiO_2_(B) (hereinafter TO) was fabricated using a similar procedure, but without Ni(NO_3_)_2_∙6H_2_O and Zn(NO_3_)_2_∙6H_2_O.

### 2.2. Characterization

Powder X-ray diffraction data (XRD) for samples were collected on a SmartLab from (Rigaku, Tokyo, Japan) and D8Advance (Bruker, Billerica, MA, USA) diffractometers using Cu*K*_α_-radiation (8048.0 eV). The crystalline phases were identified via a comparison of obtained patterns with PDF-2 (2015) cards. The lattice parameters were refined through Rietveld analysis using the JANA2006 program (ver. 25 June 2021) developed in the Institute of Physics of Academy of Sciences of the Czechia (Praha, Czechia). The morphology was studied by scanning (SEM) and scanning transmission (STEM) electron microscopy on a S5500 microscope from the Hitachi (Tokyo, Japan) equipped with a Duo-STEM detector. The elemental composition was investigated using an electron probe X-ray microanalyzer coupled to a S5500. The element distribution maps were obtained on a TM3000 microscope (Hitachi, Tokyo, Japan) with a Quantax 70 energy-dispersive X-ray spectrometer (EDX). Nitrogen adsorption/desorption isotherms were registered on an ASAP 2020 instrument (Micrometrics, Norcross, GA, USA) at 77 K. The specific surface area and pore size distribution were calculated using Brunauer–Emmett–Teller (BET) and Barrett–Joyner–Halenda (BJH) theories. The chemical state of elements was determined using X-ray photoelectron spectroscopy (XPS) on a SPECS system (SPECS, Berlin, Germany) equipped with a Phoibos-150 hemispherical energy analyzer using an Mg*K*_α_-source (1253.6 eV). The XPS binding energy scale was calibrated against the position of C 1*s* hydrocarbons (285.0 eV). The conductivity of samples was tested using the electrochemical impedance spectroscopy (EIS) technique at room temperature with a two-electrode cell on a SI1260 instrument from the Solartron Analytical (Farnborough, Hampshire, UK) over the frequency range 10^−2^–10^6^ Hz. For deeper insight into the electronic properties of materials, electronic paramagnetic resonance (EPR) and ultraviolet–visible spectroscopy (UV–Vis) analyses were performed. EPR spectra were recorded on an EMX6/1 spectrometer from the Bruker (Billerica, MA, USA) operating at X-band frequency with 100 kHz field modulation and a microwave power of 2.046 mW. UV–Vis spectra were obtained with a UV-2600 spectrophotometer (Shimadzu, Kyoto, Japan) equipped with an ISR-2600Plus integrating sphere. Barium sulfate was applied as a white reference. The magnetic properties were investigated on a SQUID magnetometer MPMS (XL) from the Quantum Design (San Diego, CA, USA) in the temperature range of 2–300 K. The applied field dependence of magnetization was measured with steps of 100 and 500 Oe in the ranges from −2000 to 2000 Oe and from ±2000 to ±10,000 Oe, respectively. The temperature dependence of magnetization was obtained under a magnetic field of 1000 Oe with a step of 2 K.

### 2.3. Electrochemical Measurements

The electrochemical parameters of TiO_2_(B) nanobelts doped with nickel and zinc were tested using two-electrode ECC-STD cells from the El-Cell GmbH (Hamburg, Germany). The working electrodes were fabricated according to the standard doctor blade technique. In brief, active material, Timcal Super P carbon black, and polyvinylidene fluoride at a weight ratio of 80:13:7 (for LIBs) or 75:18:7 (for SIBs) were dispersed in *N*-methylpyrrolidone under protracted stirring to form a slurry. The binder was placed first, whereas active material and electroconductive additive were preliminary mixed using a Pulverisette 7 planetary mill (Fritch, Idar-Oberstein, Germany). Applying a MSK-AFA-I machine (MTI Corporation, Richmond, CA, USA), the slurry was casted uniformly on a copper foil current collector pretreated with diluted hydrochloric acid. After drying at 60 °C to constant weight, working electrodes were punched, pressed on a hydraulic press at 1000 kg·cm^−2^, and heated under vacuum at 120 °C during 10 h. The active mass loading for individual electrodes was about 2 mg·cm^−2^. Assembling electrochemical cells was carried out in an argon-filled glove box with moisture and oxygen below 5 ppm. A lithium or sodium metal foil was employed as the counter or reference electrode. The electrolyte for LIBs was 1 M LiPF_6_ solution in a mixture of ethylene carbonate and diethyl carbonate at a volume ratio of 1:1 (Sigma-Aldrich, St. Louis, MO, USA). For SIBs, it was a 1 M solution of NaClO_4_ in propylene carbonate with fluoroethylene carbonate additive (5 vol.%). A 2400 membrane from the Celgard (Charlotte, NC, USA) or GF/C glass fiber from the Whattman (Little Chalfont, Buckinghamshire, UK) was used as a separator for LIBs or SIBs, respectively. The cells were tested on a Celltest System (Solartron Analytical, Farnborough, Hampshire, UK)) at room temperature in the potential range of 1–3 V (vs. Li/Li^+^) or 0.005–3 V (vs. Na/Na^+^). The performance parameters were measured by galvanostatic charge/discharge at various current densities from 30 to 1800 mA·g^−1^. Cyclic voltammetry (CV) data were collected at a sweep rate of 0.1 mV·s^−1^. The electrode impedance was measured from 10^6^ to 10^−1^ Hz after five initial CV cycles (fully desodiated) following 6 h of rest to obtain the steady-state potential.

## 3. Results and Discussion

### 3.1. Structure, Morphology, and Electronic Properties of Ni- and Zn-Doped TiO_2_(B) Nanobelts

XRD patterns of prepared materials are depicted in Figure 1. Their analysis revealed that monoclinic bronze of TiO_2_ (PDF #46-1238, space group *C*2/*m*) was a dominant phase and tetragonal anatase (PDF #21-1272, space group *I*41/*amd*) was an impurity for all samples confirming the correct synthesis [37]. Furthermore, XRD data indicated traces of monoclinic anasovit (PDF #23-0606, space group *C*2/*m*) in the products obtained due to temperature-induced TiO_2_(B)→Ti_3_O_5_ transitions [37]. No peaks of other phases were detected for TO-Ni-02, TO-Ni-05, and TO-Zn-02 samples. At the same time, NiO and ZnO were found in the compositions of TO-Ni-08 and TO-Zn-05, respectively, which were synthesized using high amounts of Ni(NO_3_)_2_∙6H_2_O and Zn(NO_3_)_2_∙6H_2_O precursors. Note that nickel oxide was formed only at the Ni/Ti ratio of 0.08, whereas zinc oxide was obtained already at Zn/Ti = 0.05. These data indicate that the solubility limit of dopant through the substitutional mechanism within titanium dioxide was reached. Hence, a further increase in dopant concentration would not be rational.

Through Rietveld refinement (Appendix A and Appendix A), the lattice parameters for unmodified titanium dioxide were calculated as *a* = 12.393(2) Å, *b* = 3.6863(9) Å, *c* = 6.505(1) Å, and *β* = 108.85(2)°. Doping with nickel changed these parameters, increasing the unit cell volume by 3–4%: from 281.23(9) Å^3^ for undoped TiO_2_(B) to 289.0(1) and 291.8(1) Å^3^ for TO-Ni-02 and TO-Ni-05 samples, respectively. This was seemingly due to the larger radius of Ni^2+^ ions (0.69 Å; here and below, these values are given according to R.D. Shannon [38] for the coordination number of 6) occupying the Ti^4+^ sites (0.605 Å). Upon a further increase in nickel content (TO-Ni-08), the volume of the unit cell decreased to 289.7(2) Å^3^, indicating a limitation in its incorporation into bronze TiO_2_ crystal structure. From refined data, the lattice parameters of TO-Ni-05 were measured as *a* = 12.269(4) Å, *b* = 3.795(1) Å, *c* = 6.601(2) Å, and *β* = 108.33(3)°. Similar to nickel, upon Zn doping, the lattice constants of TiO_2_(B) were changed. Interestingly, the unit cell expanded by about 2% for the TO-Zn-02 product (286.31(5) Å^3^). The large ionic radius of Zn^2+^ (0.74 Å) probably hindered the accentuated occupation of Ti^4+^ positions. Analogous results were previously reported for Zn-doped anatase [39]. In this way, the absence of expected ZnO reflections in the XRD pattern of TO-Zn-02 could be explained by its low content.

The SEM images unveiled in Figure 2 demonstrate that synthesized materials had the same morphology and consisted of belt-like nanostructures. The dimensions of nanobelts were 40–160 nm in width, 3–7 nm in thickness and up to a few micrometers in length. It can be seen that Ni-doped TiO_2_(B), i.e., the TO-Ni-05 sample, showed a better dispersion and uniformity of belts as compared to others.

According to the STEM study of the TO-Ni-05 product (Figure 3a), the surface of Ni-modified bronze TiO_2_ nanobelts was rough and porous. It is believed that these features may be valuable in terms of the electrochemical performance of the material as an anode for metal-ion batteries.

EDX microanalysis (Appendix A) confirmed that, along with Ti and O elements, the TO-Ni-05 product contained Ni. The Ti, O, and Ni distribution maps are depicted in Appendix A. The N_2_ adsorption/desorption data (Figure 3b) show that the TO-Ni-05 material possessed a BET specific surface area of 114 m^2^·g^−1^ and a BJH pore volume of 0.48 cm^3^·g^−1^. The pore size distribution curve (Figure 3b, inset) of nickel-doped titanium dioxide with a maximum at ∼4.2 nm shows that at least 70% of the pore volume was formed by mesopores. For the unmodified sample (Appendix A) the BET surface area and BJH pore volume were lower: 40 m^2^·g^−1^ and 0.27 cm^3^·g^−1^. This means that doping had an effect on the texture of bronze TiO_2_ nanobelts. According to the literature [40], the observed effect can be explained by the inhibition the growth of TiO_2_(B) crystallites due to nickel staying at grain boundaries and the formation Ni–O–Ti bonds.

The chemical state of elements in Ni-doped TiO_2_(B) nanobelts was investigated using the XPS method, as shown in Figure 4. According to analysis of the survey spectrum (Figure 4a), titanium, oxygen, and nickel elements existed in the TO-Ni-05 material. Furthermore, carbon is detected in the sample due to adventitious contamination. The Ti 2*p* region in Figure 4b shows a doublet at 458.4 eV (Ti 2*p*_3/2_) and 464.1 eV (Ti 2*p*_1/2_). The splitting for Ti 2*p* was equal to 5.7 eV, meaning that titanium was in a +4 oxidation state [41]. The O 1*s* spectrum could be deconvoluted into two peaks, as shown in Figure 4c. The signal at 529.4 eV could be ascribed to oxygen chemically bound to titanium, while the peak at 530.7 eV was related to adsorbed water or organics [42]. Figure 4d illustrates the spectrum of Ni 2*p*_3/2_ centered at 854.0 eV, indicating that nickel was in the +2 state [43]. Through a quantitative XPS analysis of the surface composition for the TO-Ni-05 sample, the O/(Ti+Ni) atomic ratio was calculated to be close to 1.85 (Figure 4a, inset; excluding carbon contaminations). The oxygen deficiency in nickel-doped TiO_2_(B) was probably due to neutralizing Ni^2+^ ions by oxygen vacancies in order to maintain the electrostatic balance [44], as defined in Equation (1).
(1)$$\mathrm{NiO}\overset{{\mathrm{TiO}}_{2}}{\to }{\mathrm{Ni}}_{\mathrm{Ti}}^{\prime }+V_{o}^{••}{+O}_{o}.$$

In this way, it should be noted that recently reported density functional theory calculations [36] predicted lower sodiated energy barriers for oxygen-deficient bronze titanium dioxide, presuming greater activity of Na^+^ ions during insertion and extraction into/from the framework.

In order to study the effect of doping on TiO_2_(B) electronic properties, EIS measurements were carried out for all samples. The representation of impedance spectra in Nyquist diagrams (*Z*″ vs. *Z*′) reveals depressed semicircle, reflecting the conductivity of the material and arc characterizing the interfacial phenomena, as shown in the enlarged view in Figure 5a (full-scale range EIS spectra are given in Appendix A). The analysis of impedance plots was performed in ZView 3.3c software using the electrical equivalent circuit (Figure 5a, inset), comprising the electrode resistance (*R*_el_), bulk resistance of the sample (*R*_b_) and its geometric capacitance (*C*_g_), double-layer capacitance (*C*_dl_), and charge transfer resistance (*R*_ct_). According to impedance data fitting, the conductivity of nickel-containing materials was calculated as 9.78 × 10^−10^ S·cm^−1^ (TO-Ni-02), 2.24 × 10^−8^ S·cm^−1^ (TO-Ni-05), and 5.48 × 10^−9^ S·cm^−1^ (TO-Ni-08), whereas, for the TO sample, it was 1.05 × 10^−10^ S·cm^−1^. Hence, the conductivity of titanium dioxide increased by approximately 100-fold after the incorporation of nickel dopant. In accordance with [45], this can be explained by the generation of a localized Ni 3*d* energy state within the TiO_2_(B) band structure. It is important to note that the electronic properties of titanium dioxide were improved steadily up to Ni/Ti = 0.05, while a subsequent extension in nickel content (Ni/Ti = 0.08) was accompanied by a decrease in conductivity. This was seemingly due to low-to-moderate amounts of Ni^2+^ ions being successfully incorporated into the TiO_2_(B) crystal lattice, while an excess of dopant led to the formation of an NiO phase with pronounced dielectric properties (conductivity equal to 10^−13^ S·cm^−1^ [46]). Regarding the TO-Zn-02 and TO-Zn-05 samples, it can be concluded that Zn doping also had an influence on the electroconductive properties of bronze TiO_2_. In particular, the conductivity of TO-Zn-02 and TO-Zn-05 materials was measured as 3.29 × 10^−9^ and 2.09 × 10−^9^ S·cm^−1^, respectively.

To gain insight into the electronic structure of nickel- and zinc-containing TiO_2_(B), UV–Vis experiments were performed for TO, TO-Ni-05, and TO-Zn-02 materials in the wavelength range of 200–800 nm, as shown in Figure 5b. The data obviously reveal changes in the absorbance curve, with a shift in the absorption edge to the visible-light region for nickel-containing TiO_2_(B) nanobelts. According to previous reports [45,47], this red shift may be related to the appearance of Ni 3*d* energy levels in the middle of the TiO_2_ band gap. Furthermore, the spectrum of the TO-Ni-05 sample showed an absorption peak with a maximum at approximately 725 nm, the intensity of which strongly depended on nickel concentration (Appendix A). This peak has previously been reported in the literature and was attributed to Ni^2+^ ions in the octahedral coordination [48]. With regard to Zn-doped TiO_2_(B), its absorption edge also moved slightly toward longer wavelengths (i.e., red-shifted) as compared to the undoped sample. According to the literature [47], this may be due to the introduction of impure Zn 3*d* electronic levels mixed with the O 2*p* states. The band gap energy, $E_{g}$, of materials was estimated using the Tauc method (Equation (2)) for indirect (γ = 2) electron transition via the Kubelka–Munk function $F\left(R_{\infty }\right)$.
(2)$${\left(F\left(R_{\infty }\right)\cdot h\nu \right)}^{1/\gamma }=B\left(h\nu -E_{g}\right),$$
where $F\left(R_{\infty }\right)={\left(1-R_{\infty }\right)}^{2}/2R_{\infty }$ was used instead of the absorption coefficient *α*, $R_{\infty }=10^{-A}$, *A* is the absorbance, *h* is the Planck constant, *ν* is the photon frequency, and *B* is a constant.

As revealed from measurements (Figure 5b, inset), doping with metals reduced the band gap of bronze titanium dioxide from 3.28 eV (TO) to 3.21 eV (TO-Zn-02) and 2.70 eV (TO-Ni-05).

Since EPR is an excellent technique for identifying the paramagnetic centers in materials, it was applied to study the Ni-doped TiO_2_(B) nanobelts (Figure 6a). The EPR spectrum of TO-Ni-02 powder contained a high-intensity broad signal, a narrow component on its high-field shoulder, and a low-intensity asymmetric line with the *g*-factor equal to 4.35. The *g*-factor value is a characteristic of the ions in 3*d*^5^ electron configuration in the crystal fields with a strong rhombic component [49]. It was registered for all studied samples, including TO (Appendix A), belonging to Fe^3+^ ions present in the precursor in trace amounts. Deconvolution of the TO-Ni-02 spectrum into components (designated as TO-Ni-02* in Figure 6a), excluding the contribution of the low-intensity signal of Fe^3+^, showed that the broad signal (N1) was characterized by *g* = 2.19, and the narrow one (N2) was characterized by *g* = 2.003. The *g*-factor value for component N1 is typical of Ni^2+^ ions in the octahedral crystal field [50,51]. Hence, it can be suggested that Ni^2+^ ions substituted the Ti^4+^ species in some octahedra within the TiO_2_(B) lattice. Due to this Ni^2+^/Ti^4+^ replacement, the appearance of oxygen vacancies was required to maintain charge neutrality in the crystal structure [36,52]. The value of *g*-factor for component N2 is typical of so-called *F*-centers, i.e., electrons “trapped” by the structural defects. Accordingly, they may be the conduction band electrons “trapped” with the oxygen vacancies. The EPR spectrum of TO-Ni-05, without taking into account the contributions of low-intensity signals from Fe^3+^ and *F*-centers, can be represented as a superposition of two broad signals: the N1′ component with *g* = 2.20 and the N3 contribution with *g* = 2.33 (TO-Ni-05* deconvolution in Figure 6a). As in the spectrum of the TO-Ni-02 material, the N1′ component corresponded to Ni^2+^ ions embedded in the TiO_2_(B) lattice at the Ti^4+^ positions. The N3 component, the *g*-factor value of which significantly exceeded that of isolated Ni^2+^ ions, possibly belonged to the ferromagnetically ordered structures of nickel, perhaps magnetic nickel clusters, formed from the excess dopant on its surface and/or in the pores [53]. In addition, the carrier of ferromagnetism in TO-Ni-05 can be an *F*-center-bound magnetic polaron [54,55]. In such a polaron, an electron trapped by an oxygen vacancy effectively binds the *d*-electrons of the surrounding magnetic ions. It is worth noting that, in the interpretation of the magnetic polaron model, the smaller intensity of the signal from *F*-centers in the sample synthesized with using a greater amount of nickel-containing precursor can be explained by the presence of the larger number of *F*-center-bound magnetic polarons in it.

Figure 6b represents the magnetic field dependence of magnetization under ambient conditions for the TO-Ni-05 product. The hysteresis loop for Ni-doped TiO_2_(B) nanobelts was observed, showing ferromagnetism at room temperature. At the same time, it is known that pure titanium dioxide is diamagnetic. There is currently no unified explanation for the ferromagnetic properties arising in diamagnetic materials. Many researchers have explained this fact through the model of bound polarons [54,55]. However, there are an increasing number of studies suggesting that structural defects (oxygen vacancies) have a strong effect on the magnetic behavior of TiO_2_ [55,56]. Accordingly, it seems that both mechanisms contributed to the ferromagnetism of Ni-containing TiO_2_(B). Notably, the obtained hysteresis loop was asymmetric to the origin coordinates (Figure 6b, enlarged view of the *M*(*H*) curve), indicating the existence of antiferromagnetic/ferromagnetic interactions in the analyzed sample. The presence of an antiferromagnet phase was confirmed by the temperature dependence of magnetization (Figure 6b, inset of *M*(*T*) curve), in which the peak at a temperature of 68 K inherent to transition from an antiferromagnetic state was clearly observed. Nevertheless, we were unable to refer this temperature to the Néel temperatures of known antiferromagnets containing nickel and titanium. Moreover, it should be noted that defect regions with different spin orders can form on the surface of TiO_2_ nanobelts, leading to strong exchange interactions of the antiferromagnet/ferromagnet type. Thus, to reveal the origin of shifting the hysteresis loop at a temperature of 300 K and to identify the nature of antiferromagnetic phase in Ni-doped TiO_2_(B) sample, an additional study is required.

Thus, XRD, XPS, EIS, UV–Vis, EPR, and magnetization measurements confirmed the successful substitution of Ti atoms for Ni species in the TiO_2_(B) structure, accompanied by the formation of solid solutions with general formula Ti_1−*x*_Ni*_x_*O_2−*δ*_(B). The results are in a good accordance with the material morphology (Appendix A). Indeed, the white color of unmodified bronze titanium dioxide became yellow after Ni doping. On the other hand, it was found that the zinc hardly entered into bronze TiO_2_ lattice to substitute titanium, nevertheless enhancing its electronic properties. The color of the Zn-modified sample did not change after doping.

### 3.2. Electrochemical Performance of Ni- and Zn-Doped TiO_2_(B) Nanobelts in Lithium and Sodium Batteries

Figure 7a represents the initial galvanostatic charge–discharge curves of the first cycle for TO-Ni-05, TO-Zn-02, and TiO electrodes registered in electrochemical half-cells against Li metal within the potential range of 1–3 V at the current density of 50 mA·g^−1^. The profiles look similar for tested materials, showing an identical reaction mechanism associated with lithium ion intercalation/deintercalation into/from the titanium dioxide structure according to Equation (3).
(3)$${\mathrm{TiO}}_{2}+x{\mathrm{Li}}^{+}+xe^{-}↔{\mathrm{Li}}_{x}{\mathrm{TiO}}_{2}\;\left(x\leq 1\right).$$

The initial specific capacities of unmodified TiO_2_(B) nanobelts were around 223 mAh·g^−1^ (charge) and 165 mAh·g^−1^ (discharge), showing a Coulombic efficiency of about 74%. The observed irreversibility during the first cycle (58 mAh·g^−1^) may be explained by the presence of residuals on the surface (a further discussion is provided later) and the trapping of lithium ions into irreversible TiO_2_ sites [57,58]. Furthermore, due to the presence of surface H_2_O or O–H groups, nano-TiO_2_-based anodes (as well as Li_4_Ti_5_O_12_ [59]) may demonstrate certain reactivity toward the electrolyte, causing its decomposition during the first cycle through the formation of a solid electrolyte interphase layer (SEI) [60,61]. Doping with zinc decreased the initial losses to 49 mAh·g^−1^. Indeed, during the first cycle, the TO-Zn-02 electrode gave 219 and 170 mAh·g^−1^ upon lithiation and delithiation, respectively, revealing that over 77% of initial storage was maintained. This may have been due to ZnO decreasing the surface reactivity, thereby mitigating the electrolyte decomposition upon first lithium insertion [62]. Regarding the TO-Ni-05 sample, charging and discharging capacities of 254 and 189 mAh·g^−1^ were registered. Thus, it can be suggested that Ni-doped bronze titanium dioxide revealed better lithium storage than unmodified and Zn-containing TiO_2_(B). At the same time, the TO-Ni-05 electrode possessed the greatest irreversible losses during the first cycle (65 mAh·g^−1^). This was seemingly due to its large specific surface area and high porosity.

The detected positive effect from nickel doping become more evident during continuous testing of the materials, as shown in Figure 7b. The main capacity decay for analyzed samples occurred during the initial 6–7 cycles, after which the cycling performance stabilized. This continuing degradation was associated with the loss of guest Li^+^ ions within the host TiO_2_ lattice and SEI variations. During the 10th cycle, Ni-doped TiO_2_(B) nanobelts kept around 92% of their initial reversible capacity. Upon further cycling, the average capacity decay for TO-Ni-05 was merely around 0.01 mAh·g^−1^ per cycle, which is an obvious improvement compared to TO (~0.08 mAh·g^−1^) and TO-Zn-02 (~0.04 mAh·g^−1^). The specific capacity of about 173 mAh g^−1^ could be maintained after 100 cycles for Ni-containing TiO_2_(B) nanobelts. In the case of undoped and Zn-modified materials, capacities of approximately 140 and 151 mAh·g^−1^, respectively, were achieved at the 100th cycle. Note that the results obtained for undoped TiO_2_(B) nanobelts are in good accordance with the literature [63,64].

To analyze the electrochemical Li^+^-insertion/extraction behavior of Ni-doped TiO_2_(B), CV tests (Figure 7c) were performed in the voltage range from 1 to 3 V at a scan rate of 0.1 mV·s^−1^. It can be seen that a few redox peaks existed in CV curves. A pair of intense peaks with precise splitting (so-called S-peaks [11]) at around 1.47/1.55 V (cathode region) and 1.58/1.66 V (anode region) could be attributed to lithiation and delithiation of TiO_2_(B), while the cathode and anode peaks near 1.72 and 1.98 V (A-peaks) were related to anatase. It is noteworthy that S-peaks in CVs almost completely overlapped in subsequent cycles, showing the good stability of the TiO_2_(B) lattice containing Ni dopant. The broad cathode peak between 2.06 and 2.42 V registered in the first cycle had no pair in the corresponding anode region, as well as no overlap in subsequent CVs, indicating that the irreversible nature of this process caused its appearance. According to [57], it is associated with the presence of residual water or carbon species and radicals adsorbed on the surface of titanium dioxide nanoparticles (consistent with XPS data).

By testing rate performance (Figure 7d), a reversible capacity of around 101 mAh·g^−1^ was obtained for undoped bronze titanium dioxide nanobelts at the current load of 150 mA·g^−1^, which then decreased to about 73, 55, and 11 mAh·g^−1^ at current rates of 300, 700, and 1800 mA·g^−1^, respectively. It is known that, along with simplicity of preparation, belt-like TiO_2_(B) nanostructures possess poorer rate characteristics (close to bulk TiO_2_(B) [65,66]) as compared to others (Appendix A). At the same time, when doped, bronze titanium dioxide nanobelts demonstrate an improved rate capability with higher delivered capacity. In particular, upon increasing the current density to 150, 300, 700, and 1800 mA·g^−1^, the TO-Zn-02 electrode maintained about 117, 95, 83, and 48 mAh·g^−1^. However, the best properties were detected for Ni-containing TiO_2_(B), i.e., the TO-Ni-05 sample; a reversible specific capacity of approximately 152, 136, 128, and 104 mAh·g^−1^ was achieved at rates of 150, 300, 700, and 1800 mA·g^−1^, respectively. This was seemingly due to the enhanced textural characteristics and improved conductive properties of the TO-Ni-05 product. Thus, the direct comparison in terms of cycling and rate performance of bronze titanium dioxide nanomaterials tested in this work indicated an impressive effect of Ni dopant. Hence, it is believed that the reported results may be useful for preparing advanced TiO_2_(B)-based anodes for LIBs using such effective nanostructures as nanofibers, nanowires, nanotubes, and nanosheets and/or for preparing hybrids and nanocomposites with carbonaceous materials (especially graphene). After high-rate testing, it was observed that TO-Ni-05 restored its capacity without obvious losses when the current load returned to 30 mA·g^−1^. It should be noted that, according to previous reports, the achieved rate capability results may be further enhanced through the optimization of electrode fabrication and rate test conditions, including carbon additive content and pressing options [57], type of applied binder or electrolyte composition [67], employed cutoff potentials (usually 1.2 and 2.5 V [68]), or charge procedure [69]. However, such experiments are outside the scope of this study and may be performed later.

The electrochemical performance of Ni-doped TiO_2_(B) nanobelts was next tested in SIBs, contributing to the search for materials having an open structure with a tolerable lattice distortion upon continuous sodium ion insertion/extraction. Figure 8a depicts the five initial charge/discharge voltage profiles of the TO-Ni-05 electrode, registered between cutoff potentials of 0.005 and 3 V at a current density of 35 mA·g^−1^. The data show that sodiation and desodiation capacities of around 337 and 133 mAh·g^−1^ were ascertained for the TO-Ni-05 sample during the first cycle. Hence, the initial Coulombic efficiency for the material was calculated to be about 40%, indicating huge irreversible capacity losses (204 mAh·g^−1^), in accordance with other TiO_2_-based anode materials for SIBs. According to a literature review [70,71,72], the main reasons for this observed irreversibility are (i) side reactions at the electrode/electrolyte interface (usually more intense for mesoporous materials with a relatively high specific surface area, such as TO-Ni-05), (ii) trapping of some Na^+^ ions within the TiO_2_(B) lattice, (iii) amorphization of anatase due to interactions with sodium, and (iv) sodiation of the comprised conductive carbon. Meanwhile, an analysis of subsequent voltage curves demonstrated that they almost overlapped after the third cycle, presenting good reversibility following the electrochemical storage process.

Figure 8b compares the cycling performance of unmodified and Ni-doped bronze TiO_2_ nanobelts. It can be observed that TO-Ni-05 material maintained a capacity of around 97 mAh·g^−1^ after 50 cycles with a Coulombic efficiency of ~99.3%. This is much higher than the corresponding values of 59 mAh·g^−1^ and 98.8% observed for the TO electrode for the same duration of cycling. Retentions of nearly 73% (TO-Ni-05 sample) and 52% (TO product) of the initial desodiation capacity were observed after 50 charge/discharge tests, revealing more stable Na^+^ insertion/extraction into/from Ni-modified titanium dioxide.

Figure 8c displays the CV curves of the TO-Ni-05 electrode measured in the potential range of 0.005–3 V (vs. Na/Na^+^). Obviously, the current response during the first cathodic sweep was much higher than that in subsequent scans, confirming the occurrence of irreversible reactions associated with electrolyte reduction (around 1.01 V), decomposition of organics, formation of amorphous phase(s), and trapping of Na^+^ ions [71,73]. Furthermore, the intensive peak at the end of the cycle may have involved the sodiation of carbon in the electrode. Regarding the anode side, it should be noted that, from the second CV onward, the broad peak at 0.4–0.9 V became pronounced. According to the literature, this may be related to Na^+^ extraction from TiO_2_(B) [74]. Upon subsequent cycling, the current of the peak increased, assuming activation of the structure. On the other hand, no distinguishable cathodic feature was detected in the second or subsequent cyclic sweeps. A similar phenomenon was previously observed in a study on nanoparticulate TiO_2_(B) [73]. It was established that Na^+^ insertion into bronze titanium dioxide occurs in a wide interval of potentials (mainly below 1 V) [22,27]. Lastly, contrary to recent results [73,75], no characteristic anode peak was registered for conductive carbon desodiation, suggesting its negligible effect on the overall storage capacity of the TO-Ni-05 electrode.

In order to study the relationship among battery performance, kinetic characteristics, and diffusion phenomenon, the EIS spectrum of TO-Ni-05 electrode was recorded at the end (fully desodiated) of the fifth CV cycle (after a 5 h rest). Three main contributions can be observed in the Nyquist plot (Figure 8d): interfacial effects on the SEI layer, migration of charge carriers through the double layer at the electrode/electrolyte surface, and mass transport within the solid phase, expressed by the deformed semicircle in the high-to-medium-frequency region and low-frequency sloping line. Further analysis of the impedance spectrum was performed using the standard equivalent circuit model, as presented in the inset of Figure 8d. *R*_s_, *R*_f_, *CPE*_f_, *R*_ct_, *CPE*_dl_, and *Z*_w_ in the circuit denote the resistance of the cell (electrolyte, separator, and electrodes), the resistance and capacitance of the interfacial layer, charge transfer resistance, double-layer capacitance, and Warburg impedance. The collected data (Appendix A) show that *R*_ct_ for the TO-Ni-05 sample was equal to approximately 104 Ω, indicating its appropriate electronic and ionic conductivity. To further evaluate the effects of Ni^2+^ ions embedding into the bronze titanium dioxide lattice, the diffusion coefficient of electroactive species $D_{\mathrm{Na}}$ (cm^2^·s^−1^) was determined using Equation (4).
(4)$$D_{\mathrm{Na}}=\frac{R^{2}T^{2}}{2S^{2}n^{4}F^{4}C_{\mathrm{Na}}^{2}\sigma_{W}^{2}},$$
where *R* is the gas constant (8.314 J·mol^−1^·K^−1^), *T* is the absolute temperature (298.15 K), *S* is the geometric area of the electrode (0.5 cm^2^), *n* is the number of electrons transferred during the reaction presented in Equation (5), *F* is the Faraday constant (96,500 C·mol^−1^), *C*_Na_ is the concentration of Na^+^ species in the material (17 × 10^−3^ mol·cm^−3^), and *σ*_W_ is the Warburg factor, which can be determined from Equation (6) through linear fitting of the *Z*′ vs. *ω*^−1/2^ plot (Appendix A).
(5)$${\mathrm{TiO}}_{2}+{\mathrm{Na}}^{+}+e^{-}↔{\mathrm{NaTiO}}_{2},$$
(6)$$Z^{\prime }=R_{s}+R_{\mathrm{ct}}+\sigma_{W}\omega^{-1/2},$$
where *Z*′ is the real part of the impedance (Ω), and *ω* is the angular frequency (s^−1^).

According to the calculations, the chemical diffusion coefficient of sodium ions for the Ni-doped sample was 6.84 × 10^−13^ cm^2^·s^−1^. Using the same methodology (i.e., EIS) of *D*_Na_ determination, this value is better than that obtained for previously reported TiO_2_ materials, including anatase/C nanoparticles (1.56 × 10^−14^ cm^2^·s^−1^) [76], rutile TiO_2_ nanoparticles (9.9 × 10^−15^ cm^2^·s^−1^) [77], and anatase/C nanofibers (2.0 × 10^−13^ cm^2^·s^−1^) [78]. Furthermore, the calculated *D*_Na_ is comparable with that of other modified titanium-based oxides for SIB anodes, as shown in Appendix A. According to the abovementioned data, it is reasonable to expect that the *D*_Na_ of bronze titanium dioxide is increased after Ni doping.

## 4. Conclusions

In summary, herein, a hydrothermal route was applied to synthesize mesoporous belt-like TiO_2_(B) nanostructures doped with nickel (Ni/Ti atomic ratios of 0.02, 0.05, and 0.08) and zinc (Zn/Ti = 0.02 and 0.05) with a specific surface area and pore volume reaching 114 m^2^·g^−1^ and 0.48 cm^3^·g^−1^. According to the analysis of XRD data, nickel doping increased the unit cell volume of bronze titanium dioxide by 4% (Ni/Ti = 0.05), confirming the incorporation of Ni^2+^ ions at the Ti^4+^ positions with the formation of a substitutional solid solution. Indeed, the Ni^2+^ ion is bigger (0.69 Å) than Ti^4+^ (0.605 Å), resulting in lattice distortions after substitution. The generation of localized Ni 3*d* defect states within the band gap of TiO_2_(B) was confirmed by UV–Vis studies, revealing that the band gap energy was reduced from 3.28 to 2.70 eV after doping. XPS studies revealed an oxygen deficiency for Ni-doped TiO_2_(B). According to EPR, *F*-centers, which may represent the conduction band electrons “trapped” with oxygen vacancies, exist in the nickel-containing material. On the other hand, due to its large ionic radius, zinc (0.74 Å) hardly entered into TiO_2_(B) crystal structure, nevertheless provoking a band gap narrowing effect (to 3.21 eV). The EIS technique revealed that the conductivity of nickel- and zinc-containing titanium dioxide increased to 2.24 × 10^−8^ S·cm^−1^ (Ni/Ti = 0.05) and 3.29 × 10^−9^ S·cm^−1^ (Zn/Ti = 0.02), exceeding that of the undoped sample (1.05 × 10^−10^ S·cm^−1^). The galvanostatic charge/discharge cycling of materials in lithium cells showed a favorable effect of nickel and zinc doping on the reversibility of the electrochemical process. Among the tested samples, Ni-containing TiO_2_(B) nanobelts with an Ni/Ti atomic ratio of 0.05 demonstrated the best battery performance. In particular, after 100 charge/discharge cycles, a reversible capacity of 173 mAh·g^−1^ was achieved for nickel-doped TiO_2_(B) at the current density of 50 mA·g^−1^, whereas unmodified and zinc-doped bronze TiO_2_ electrodes maintained 140 and 151 mAh·g^−1^. Moreover, Ni doping improved the rate performance of TiO_2_(B) nanobelts. Concerning its operation in sodium cells, it was found that nickel-containing material exhibited improved cycling with a specific capacity of about 97 mAh·g^−1^ after 50 cycles at the current load of 35 mA·g^−1^. EIS studies identified a *D*_Na_ of 6.84 × 10^−13^ cm^2^·s^−1^ for Ni-doped TiO_2_(B) (Ni/Ti = 0.05). The main factors determining the enhanced electrochemical performance of doped TiO_2_(B) were (i) increased electronic conductivity, (ii) improved stability of crystal lattice toward guest ion insertion/extraction, and (ii) facilitated transport of Li^+^ and Na^+^. Thus, the current study demonstrates that proper doping might be an effective way to adopt bronze titanium dioxide’s properties for its usage in the area of metal-ion batteries.

## Figures and Tables

**Figure 1 nanomaterials-11-01703-f001:**
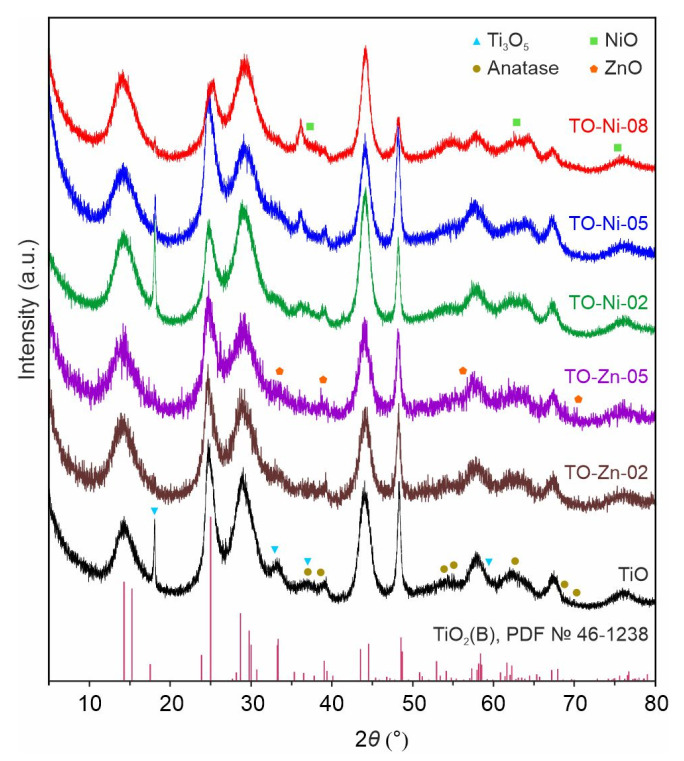
XRD patterns for as-synthesized samples doped with nickel and zinc TiO_2_(B).

**Figure 2 nanomaterials-11-01703-f002:**
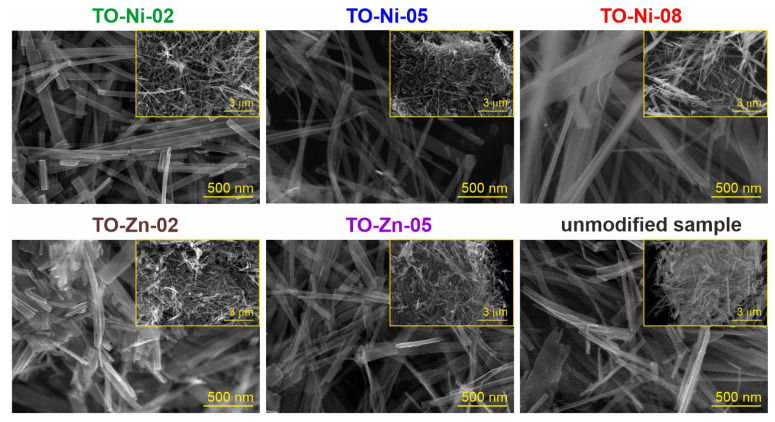
SEM images for titanium dioxide doped with nickel and zinc.

**Figure 3 nanomaterials-11-01703-f003:**
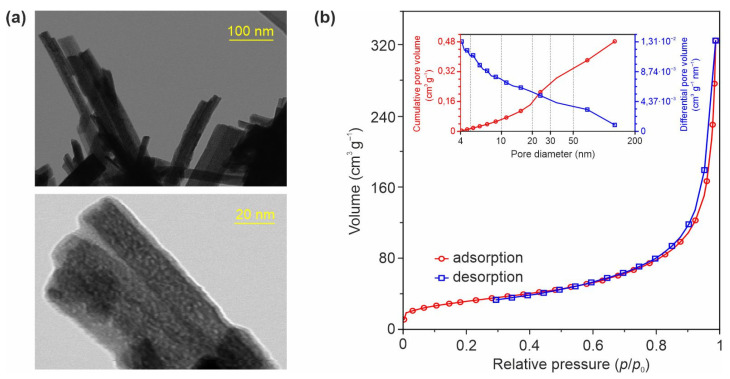
(**a**) Microphotographs in STEM mode at different magnification, (**b**) nitrogen adsorption/desorption isotherms at 77 K, and (inset) cumulative pore volume and pore size distribution curves for TO-Ni-05 material.

**Figure 4 nanomaterials-11-01703-f004:**
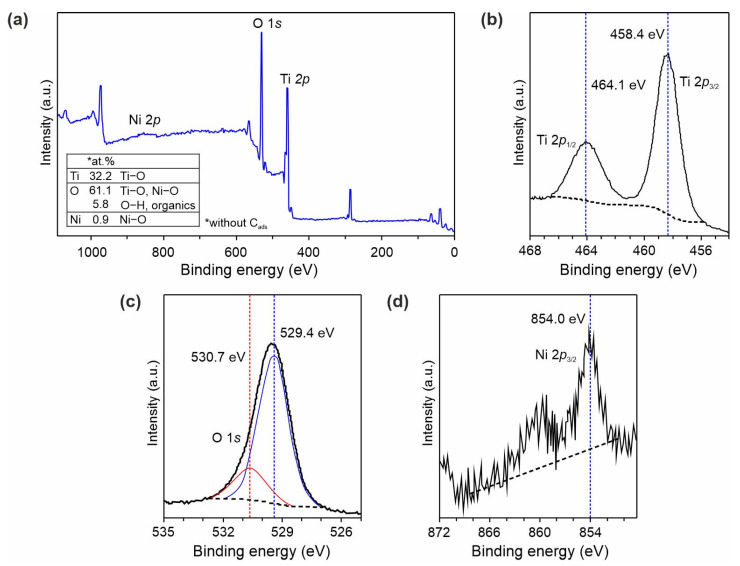
(**a**) XPS survey scan and high-resolution spectra of (**b**) Ti 2*p*, (**c**) O 1*s*, and (**d**) Ni 2*p* core levels for TO-Ni-05 sample.

**Figure 5 nanomaterials-11-01703-f005:**
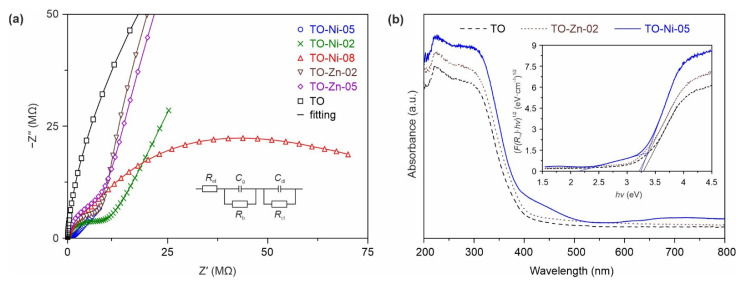
(**a**) EIS spectra (enlarged view) of TiO_2_(B) materials containing different amounts of nickel and zinc and electrical equivalent circuit (inset) for fitting of the experimental impedance data. Symbols represent experiments, whereas lines are fitting results. (**b**) UV–Vis absorption data for TO, TO-Ni-05, and TO-Zn-02 products (inset represents corresponding Kubelka–Munk plots).

**Figure 6 nanomaterials-11-01703-f006:**
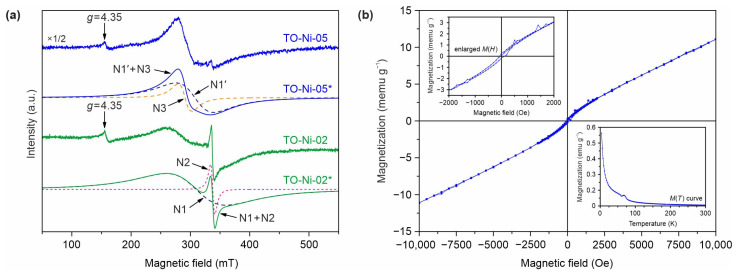
(**a**) Experimental and deconvoluted (marked by *) EPR spectra for TO-Ni-02 and TO-Ni-05 samples at 290 K. (**b**) Magnetic hysteresis loop at room temperature for TO-Ni-05 material. The insets show the enlarged view of the *M*(*H*) curve in a low magnetic field and the temperature dependence of magnetization.

**Figure 7 nanomaterials-11-01703-f007:**
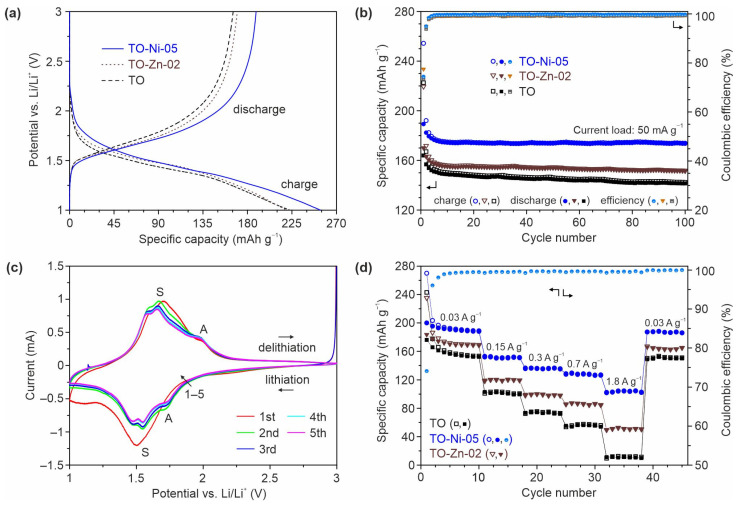
Lithium-ion battery performance of Ni- and Zn-doped bronze titanium dioxide. Charge/discharge curves of the first cycle (**a**) and cyclability (**b**) at the current load of 50 mA·g^−1^ for unmodified, Ni-doped, and Zn-doped TiO_2_(B) materials. CV data for TO-Ni-05 at a scan rate of 0.1 mV·s^−1^ (**c**). Rate performances at different current densities of 30, 150, 300, 700, and 1800 mA·g^−1^ (**d**) for TO, TO-Ni-05, and TO-Zn-02 electrodes.

**Figure 8 nanomaterials-11-01703-f008:**
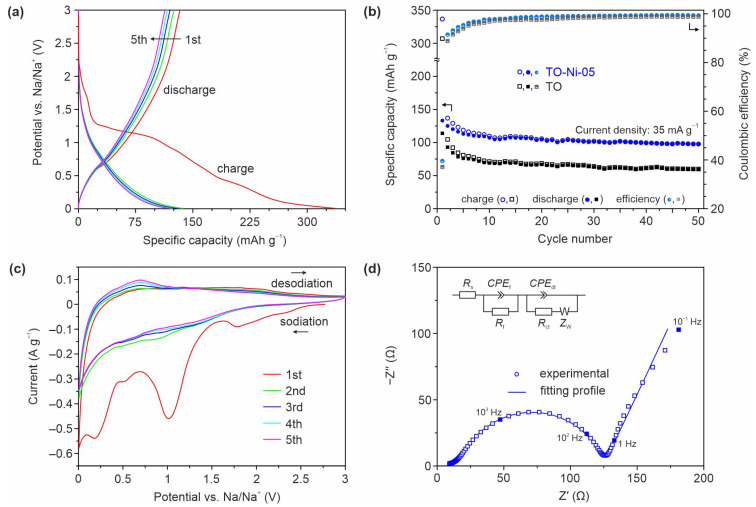
Sodium storage characteristics of analyzed TiO_2_(B) products. The five initial voltage profiles at 35 mA·g^−1^ for the TO-Ni-05 material (**a**) and the corresponding correlation of its specific capacity with cycle number as compared to that for unmodified titanium dioxide (**b**). CV curves for Ni-doped TiO_2_(B) registered at a scan rate of 0.1 mV·s^−1^ (**c**). Nyquist diagram for TO-Ni-05 electrode after five initial CV cycles and electrical equivalent circuit (inset) used for fitting the experimental data (**d**).

## Data Availability

The data presented in this study are available on request from the corresponding author.

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
