# Peer review of "Enhancing Lithium and Sodium Storage Properties of TiO2(B) Nanobelts by Doping with Nickel and Zinc"

_nanomaterials, 2021, doi:10.3390/nano11071703_

Round 1
Reviewer 1 Report
The work is excellently done, and its relevance and novelty are beyond doubt. The development of new materials for efficient energy storage devices is a significant scientific challenge.
The article contains all the necessary data to confirm the obtained results.
The work can be published after correcting some spellings.
Author Response
Reviewer 1: The work is excellently done, and its relevance and novelty are beyond doubt. The development of new materials for efficient energy storage devices is a significant scientific challenge. The article contains all the necessary data to confirm the obtained results. The work can be published after correcting some spellings.
Answer: We thank the Reviewer 1 for his/her high estimation of our paper. We have checked the manuscript carefully and tried to avoid any errors. The revised manuscript was reviewed by our colleague who is well-versed in English.
Reviewer 2 Report
This manuscript reported the nickel- and zinc-doped TiO2(B) electrodes and their electrochemical properties for Li and Na ion storage. In my view, this manuscript can be recommended for publication in Nanomaterials if the authors address the following questions well.
(1) In the part of electrochemical measurements, the areal mass loading and the total mass of one electrode are missing.
(2) The equations (3), (4) and (6) have the wrong information.
(3) Is there any big difference in Nyquist plots for the TO-Ni-05 electrode before cycles and after five cycles?
(4) The authors are suggested to add a table in supporting information to show that the chemical diffusion coefficient of sodium ions for Ni-doped is better than for previously reported electrode materials.
(5) There are still some grammatical errors, especially singular and plural errors.
(6) Is it necessary to use 85 references for a research article instead of a review article?
Author Response
Reviewer 2: This manuscript reported the nickel- and zinc-doped TiO2(B) electrodes and their electrochemical properties for Li and Na ion storage. In my view, this manuscript can be recommended for publication in Nanomaterials if the authors address the following questions well.
Answer: We thank the Reviewer 2 for his/her constructive criticism and comments, which allowed us to improve the manuscript. Our point-by-point response is presented in the attachment below.

Reviewer 3 Report
General comments Well written manuscript. Needs minor revisions. Specific comments Introduction is weak. There is an abrupt jump from last sentence of first paragraph which says " Therefore, searching the materials with acceptable stability of structure........" to first sentence of second paragraph "Against this background, researchers have paid heightened attention to titanium dioxide". Authors must justify why they used TiO2, when many transition metal oxides perform better in NIB and LIB compared to it. Authors say " At the same time, due to the inability of stable Na+ insertion/extraction into/from graphite, the using of TiO2(B) as an anode material for SIBs is being studied recently " but do not give any references. Despite its 'perceived' advantages, TiO2 shows extensive lithium dendrite growth which limits ts utility in LIB and NIB. Authors must mention a thing or two about dendrite problem. Authors must justify why limit the level of doping only to 0.02, 0.05, and 0.08. Why not higher? Results and Discussion is written well except for some misleading statements like: "uniform distribution of nickel in the material can be viewed.." which is not true. Both dispersion and dispersion is not good. Modify your statement. Authors must explain the presence of substantial carbon peak in XPS. Rectify the error in Equation 3 and 4. Some Chinese characters are appearing.Author Response
Reviewer 3: General comments Well written manuscript. Needs minor revisions.
Answer: We thank the Reviewer 3 for his/her positive opinion as well as critical comments, with help of which we were able to improve the article. The information about changes is listed in the attachment below.

Round 2
Reviewer 2 Report
The authors have revised the manuscript and well answered my questions in the revised version. I think the revised version can be recommended to be published in Nanomaterials.